# The SCC*mec* Types and Antimicrobial Resistance among Methicillin-Resistant *Staphylococcus* Species Isolated from Dogs with Superficial Pyoderma

**DOI:** 10.3390/vetsci8050085

**Published:** 2021-05-13

**Authors:** Yuttana Chanayat, Areerath Akatvipat, Jeff B. Bender, Veerasak Punyapornwithaya, Tongkorn Meeyam, Usanee Anukool, Duangporn Pichpol

**Affiliations:** 1Master’s Degree Program in Veterinary Science, Faculty of Veterinary Medicine, Chiang Mai University, Chiang Mai 50100, Thailand; yuttana.c@cmu.ac.th; 2Department of Companion Animals and Wildlife Clinic, Faculty of Veterinary Medicine, Chiang Mai University, Chiang Mai 50100, Thailand; areerath.a@cmu.ac.th; 3College of Veterinary Medicine and the School of Public Health, University of Minnesota, Minneapolis, MN 55455, USA; bende002@umn.edu; 4Veterinary Public Health and Food Safety Centre for Asia Pacific, Faculty of Veterinary Medicine, Chiang Mai University, Chiang Mai 50100, Thailand; veerasak.p@cmu.ac.th (V.P.); tongkorn.meeyam@cmu.ac.th (T.M.); 5Department of Veterinary Biosciences and Veterinary Public Health, Faculty of Veterinary Medicine, Chiang Mai University, Chiang Mai 50100, Thailand; 6Division of Clinical Microbiology, Department of Medical Technology, Faculty of Associated Medical Sciences, Chiang Mai University, Chiang Mai 50200, Thailand; usanee.anukool@cmu.ac.th; 7Infectious Diseases Research Unit (IDRU), Faculty of Associated Medical Sciences, Chiang Mai University, Chiang Mai 50200, Thailand

**Keywords:** pyoderma, type V SCC*mec*, skin swab, MRS, staphylococcal infection, zoonotic disease

## Abstract

This study characterizes clinical methicillin-resistant staphylococcal (MRS) isolates obtained from superficial pyoderma infections in dogs. Our interest was to determine the staphylococcal cassette chromosome *mec* (SCC*mec*) type and the antimicrobial susceptibility among MRS isolates from clinical cases. Skin swabs were collected and cultured. *Staphylococcus* species were identified and characterized with biochemical tests and MALDI-TOF-MS and antimicrobial susceptibility testing by disk diffusion. *mecA* detection and staphylococcal cassette chromosome *mec* (SCC*mec*) typing were achieved by PCR. Of the 65 clinical samples, 56 (86.2%) staphylococcal infections were identified. Twelve (21%) of 56 isolates were MRS infections. All MRS isolates were multidrug resistant. The *ccrC* and class-C2 *mec*, which were SCC*mec* type V, were the most prevalent (66.7%) among the 12 MRS isolates. The predominant SCC*mec* type V was found in *S. aureus*, *S. intermedius* group, *S. lentus*, *S. xylosus,* and *S. arlettae*. Treatment failure is a concern with the emergence of highly resistant MRS in dogs associated with superficial pyoderma. The detection of type V SCC*mec* MRS has previously been reported among veterinarians and dog owners but not in Northern Thailand. These infections serve as a reminder to improve infection prevention and control measures including reducing environmental contamination and potential zoonotic exposures to MRS. In addition, educational awareness of these risks in small animal hospitals needs to be increased among veterinary hospital staff, clients, and patients.

## 1. Introduction

Infection with methicillin-resistant staphylococci (MRS) is an ongoing and emerging health concern among both humans and animals. Methicillin-resistant *Staphylococcus aureus* (MRSA) not only causes hospital-, community-, and livestock-associated infections in humans, but it can also lead to infections in pet animals [1]. Methicillin-resistant *Staphylococcus pseudintermedius*, which belongs to the *S. intermedius* group (MRSIG), can cause serious wound infections in domestic companion animals such as dogs and cats. It may also be capable of spreading between animals and humans, including veterinarians, companion animal owners, and veterinary nurses [2,3]. Methicillin-resistant coagulase-negative staphylococci can persist in the patient environment and serve as a source of human infection in hospital settings [4]. MRS is recognized as a public health threat and is a significant opportunistic pathogen in both human and veterinary medicine [5].

The evolution of MRS emerged from the development of the penicillin-binding protein 2a (PBP2a), which is encoded by the *mecA* gene [1]. This protein has a significantly lower affinity for beta-lactams and thus cell wall synthesis can continue even in the presence of normally inhibitory concentrations of beta-lactam antibiotics. Thus, the detection of *mecA* by polymerase chain reaction (PCR) is used to characterize MRS [2,5]. The *mecA* gene is the mobile genetic element located in the chromosome of *Staphylococcus* known as the staphylococcal cassette chromosome *mec,* or SCC*mec* [6]. There are four important components of the SCC*mec* element: (1) the group comprised of the *mec* genes; the *mecA* gene and the regulatory genes, (2) the group comprised of the *ccr* genes responsible for the mobility of the SCC*mec* element, (3) the direct repeated nucleotide at the end of both sides of SCC*mec* making the movable structure of the SCC*mec* element, and (4) the 3′ ending part of the *orfX* gene [7]. Currently, SCC*mec* is classified into 13 types, type I to type XIII, according to the differences of the *mec* gene and the *ccr* gene [8]. A multiplex PCR is used to determine the structure of the *mec* complex, while the presence of different *ccr* genes is a popular method for SCC*mec* typing [9]. Staphylococcal infections in companion animals, particularly in dogs and cats, include superficial pyoderma, otitis externa, and superficial bacterial folliculitis. Occasionally, MRS is also detected among companion animal owners. These are mostly multidrug-resistant MRS infections [10,11]. According to prior reports, the identification of various MRS isolates, such as MRSA, MRSIG, and MRCNS, are rarely reported in Thailand. The objectives of this study were to characterize the type of staphylococcal infections associated with superficial pyoderma among dogs at the Small Animal Teaching Hospital, Chiang Mai University, Thailand.

## 2. Materials and Methods

### 2.1. Data and Sample Collection

Cases of superficial pyoderma in dogs were identified by clinical staff at our teaching hospital from 2015 to 2017. Sixty-five dogs presenting with superficial pyoderma at the Small Animal Teaching Hospital, Faculty of Veterinary Medicine, Chiang Mai University, Thailand were swabbed and submitted to the veterinary diagnostic laboratory for microbiological analysis.

### 2.2. Bacterial Isolation and Identification

The skin swabs were kept in Stuart transport medium (BD BBLTM, US) and submitted to the microbiology laboratory within 4–6 h for bacterial identification and antimicrobial susceptibility testing. Standard bacterial isolation procedures were used as follows. Staphylococcal isolates were identified in cultures using 5% of sheep blood agar (Merck, Darmstadt, Germany) along with a biochemical testing process that employed Gram stain, catalase, tube coagulase, Voges–Proskauer (VP) test, and the clumping factor [12]. All staphylococcal species were confirmed by MALDI-TOF-MS analysis.

### 2.3. Antimicrobial Susceptibility Testing

Antimicrobial susceptibility testing was performed by the disk diffusion method according to the Clinical and Laboratory Standards Institute (CLSI) recommendations [13]. Sixteen antimicrobial agents, including penicillin, ampicillin, cefoxitin, amoxicillin/clavulanate, cefazolin, cefpodoxime, amikacin, gentamicin, doxycycline, ciprofloxacin, chloramphenicol, clindamycin, erythromycin, linezolid, rifampin, and trimethoprim/sulfamethoxazole, were used in this study. The minimum inhibitory concentration (MIC) of oxacillin was determined for all *Staphylococcus* isolates by Vitek 2 system. All MRS isolates were identified using the CLSI oxacillin MIC breakpoints: MIC ≥ 4 µg/mL for *S. aureus* and MIC ≥ 0.5 µg/mL for other *Staphylococcus* spp. [13]. *Staphylococcus aureus* ATCC 25923 was used as a quality control strain.

### 2.4. PCR Detection of mecA Gene

DNA was extracted using a commercial test kit for genomic DNA obtained from tissue samples (NucleoSpin R Tissue). The *mecA* genotype of staphylococci was characterized. The 50-µL PCR reaction mixtures contained 100 ng chromosomal DNA, oligonucleotide primers (10 pmols), 2.5 mM each deoxynucleotide triphosphates, Taq buffer, and 5 U Taq polymerase (I-TagTM plus iNtRON Biotechnology, Korea) at a final volume of 50 µL. A PCR thermal cycler was used for amplification with an initial denaturation step (94 °C, 5 min) comprised of 40 cycles of denaturation (95 °C, 45 s), an annealing step (59 °C, 45 s), and an extension step (72 °C, 45 s); and a final extension step at 72 °C for 5 min. Regarding the primer pairs used for PCR experiments, the forward primer was 5′-TGGCTATCGTGTCACAATCG-3′, the reverse primer was 5′-CTGGAACTTGT TGAGCAGAG-3′, and the product size was 309 base pairs [14].

### 2.5. SCCmec Typing

The SCC*mec* typing of all *mecA* gene-positive staphylococcal isolates was performed using the multiplex PCR, M-PCR1, and MPCR2, as previously described by [15]. The M50-µL reaction mixtures of M-PCR1 contained 100 ng chromosomal DNA, oligonucleotide primers (0.1 µM), 200 µM each deoxynucleotide triphosphates, Taq buffer, and 2.5 U DreamTaq DNA polymerase (Thermo Fisher Scientific UAB, Vilnius, Lithuania) at a final volume of 50 µL. The concentration of MgCl_2_ was 3.2 mM. A PCR thermal cycler was used for amplification with an initial denaturation step (94 °C, 2 min) comprised of 30 cycles of denaturation (94 °C, 2 min), an annealing step (57 °C, 2 min), and an extension step (72 °C, 2 min); and a final extension step at 72 °C for 2 min. The 50-µL reaction mixture of M-PCR2 contained the same components as the M-PCR1 except that the concentration of MgCl_2_ was 2 mM and the annealing temperature was raised to 60 °C for 2 min. The primer pairs used for PCR experiments are listed in Appendix A. The M-PCR 1 for *ccr* type assignment contained two primers used to detect *mecA* and eight primers used for the identification of five *ccr* genes: four primers including a common forward primer (common to *ccrB1–3*) and three reverse primers specific for *ccrA1*, *ccrA2,* and *ccrA3* used to identify *ccr1–3* based on the differences present in the *ccrA* genes; two primers used to identify *ccr4*; and two primers used to identify *ccr5*. Furthermore, the M-PCR2 for *mec* class assignment contained four primers that were used to identify the gene lineages of *mecA–mecI* (class A *mec*), *mecA*-IS*1272* (class B *mec*) and *mec*AIS*431* (class C *mec*). The positive control strains for SCC*mec* typing were 4 MRSA strains, including epidemic MRSA (EMRSA)-8 (SCC*mec* type I), N315 (SCC*mec* type II), EMRSA-4 (SCC*mec* type III), and EMRSA-10 (SCC*mec* type IV).

### 2.6. Statistical Analysis

Percentages of antimicrobial susceptibility were calculated for staphylococci and MRS isolates. The antimicrobial susceptibility was categorized as 3 groups, including susceptible, intermediate, or resistant, according to the breakpoint of the inhibition zone or MIC as recommended by CLSI M100-S26 guidelines [13]. Data were visualized and analyzed using R version 4.0.3 [16].

## 3. Results

### 3.1. Staphylococcal Infections in Dogs with Superficial Pyoderma

Staphylococci were isolated from 56 (86.2%) of 65 clinical samples. Two isolates (3.1%) were identified as *Staphylococcus aureus* and 50 isolates (76.9%) as the *S. intermedius* group. In addition, one isolate each of coagulase-negative staphylococci, *S. lentus**, S. xylosus,*
*S. lugdunensis**,* and *S. arlettae,* were found. For nine culture-negative samples, non-*Staphylococcus* bacteria were detected including beta-hemolytic *Streptococcus* Group C, *Escherichia coli*, *Aerococcus viridans*, *Proteus mirabilis*, *Pseudomonas aeruginosa*, *Rothia nasimurium*, *Klebsiella pneumoniae*, *Moraxella* sp., *Corynebacterium auriscanis*, and *Enterococcus faecalis*. From available information, the mean age of the patients was 6.2 years of age (range from 0.5 to 14 years). Twenty-eight dogs were intact males, 20 were female, and one was a spayed female. Eight (16%) dogs presented with papules, 19 (34%) presented with pustules, 35 (63%) presented with crusts, and 20 (36%) presented with epidermal collarettes. Eleven patients had recurring pyoderma and 37 had other underlying medical conditions such as atopic dermatitis, food allergies, demodicosis, hyperadrenocorticism, or hypothyroidism. Fifteen (31%) of 49 dogs with reviewable records were previously treated with antibiotics (cephalexin (*n* = 8), amoxicillin–clavulanate (*n* = 4), doxycycline (*n* = 3).

### 3.2. Antimicrobial Susceptibility Testing Results

Most staphylococci were found to be resistant to penicillin and ampicillin. All staphylococci were susceptible to linezolid and rifampin (Table 1). Twelve staphylococcal isolates were resistant to oxacillin and cefoxitin and were confirmed as MRS isolates. One isolate was *S. aureus*, eight isolates were in the *S. intermedius* group, and three isolates were identified as coagulase-negative staphylococci: *S. lentus, S. xylosus, and S. arlettae*. All MRS isolates were resistant to penicillin, cephalosporin, and fluoroquinolone. Notably, the *mec*A gene was found to be present in all methicillin-resistant staphylococci (Figure 1). Epidemiologic data were available for 10 dogs. The mean age of these 10 dogs was 6.3 years of age (ranging from 1 to 13 years). Six of the dogs were intact males and four were female dogs. Two (20%) dogs presented with papules, four (40%) presented with pustules, eight (80%) presented with crusts, and four (40%) presented with epidermal collarettes. Five of ten dogs had recurring pyoderma and five had underlying medical conditions. Furthermore, eight of the ten subjects had previously been prescribed antibiotics.

### 3.3. SCCmec Types of Methicillin-Resistant Staphylococcal Isolates

There were 12 isolates of MRS that were SCC*mec* typed using the multiplex PCR. M-PCR1 successfully amplified DNA fragments corresponding in size to each ccr gene (Figure 2) and M-PCR2 was applied to assign the *mec* class (Figure 3). Characterization results from all 12 MRS isolates revealed that ccrC and class-C2 *mec* were found from one isolate (8.3%) of *S. aureus*; thus, it could be classified as SCC*mec* type V. These components were also found in four isolates (33.3%) of the *S. intermedius* group. The ccrA1B1, ccrC, and class-C2 *mec* were found in four isolates (33.3%) of the *S. intermedius* group and they were categorized as SCC*mec* non-typeable strains. The coagulase-negative staphylococci, including *S. lentus, S. xylosus, and S. arlettae*, were found to carry ccrC and class-C2 *mec*. These strains were classified as SCC*mec* type V (Table 2).

## 4. Discussion

This study represents, to our knowledge, the first molecular characterization of MRS isolates obtained from canines with superficial pyoderma in Northern Thailand. The presence of *mecA* gene encoding PBP2a in all MRS isolates was confirmed along with the antimicrobial susceptibility profiles highlighting potential treatment failures with commonly used antimicrobials [17]. One MRSA isolate was recovered from a dog with superficial pyoderma. This emphasizes the need for clinical awareness of antimicrobial-resistant organisms and potential treatment failures and the need for optimum infection control measures in clinical settings to prevent healthcare-associated infections. Direct contact with an infected animal can be a risk for human pet owners, especially those with underlying health conditions [18]. Dogs are at risk for carrying MRS and MRSA and can be colonized with resistant strains without showing clinical signs [10,19,20]. Transmission can occur through close contact [21,22]. These findings highlight the need for veterinarian awareness of potential multidrug-resistant infections, and the importance of providing client information about the management and care of patients and the appropriate hand hygiene practices to reduce the transmission risk of multidrug-resistant infections between animals and humans.

Staphylococcal infections in domestic companion animals, especially in dogs and cats, can commonly cause superficial pyoderma, otitis externa, superficial bacterial folliculitis, and bacterial rhinitis. The main cause of these infections are often *S. pseudintermedius* [10,23,24]. The *S. intermedius* group (SIG) accounted for the majority of the isolates recovered from clinical pyoderma cases in this study. These SIG isolates from dogs with pyoderma could not be conclusively identified in the absence of molecular testing and they were indicated as *S. pseudintermedius*/*S. intermedius*. Subsequent analysis of two strains by MALDI-TOF-MS systems (VITEK MS, bioMérieux, Marcy l’Etoile France) gave the same results. Although *S. pseudintermedius* has been recognized as a major pathogen among SIG isolates, the multiplex PCR method for species identification of coagulase-positive staphylococci by targeting the *nuc* gene locus [25] should be performed. Currently, the concept of SIG is becoming more and more broad. Recently, two species, *S. cornubiensis* and *S. ursi*, have been added to the SIG [26,27]; however, they were not present in the dogs tested here. There are some reports which document MRS infection in companion animals, their owners, veterinarians, and other animal caretakers [22,28,29,30,31]. MRS can readily spread and can contaminate the immediate environment, including surfaces in veterinary clinics [32].

Previous studies have documented the increasing frequency of the recovery of MRS in canine pyoderma [33,34]. These multidrug-resistant infections identified in companion animals are difficult to treat [2,35]. For our study, MRS isolates were resistant to fluoroquinolones. The International Society for Companion Animal Infectious Diseases (ISCAID) has devised some important diagnostic and treatment approaches [36]. It is, therefore, necessary and important for Thailand and other South East Asian countries with limited culture and epidemiologic data to recognize the emergence of resistant staphylococcal infections and to follow appropriate antimicrobial stewardship principles. Awareness of the emergence of MRS is important to veterinary practitioners and future veterinarians. Due to the limited effectiveness of antimicrobials to treat MRS, good hygienic and infection control practices are needed to prevent potential environmental contamination and spread to other animals and humans.

Our study also documented that staphylococcal isolates contained SCC*mec* types V. This SCC*mec* type has previously been found in dogs living in the central part of Thailand [10]. However, these MRS strains were isolated by swabbing the nares and perineum of healthy dogs and not from the dogs with pyoderma. In addition, SCC*mec* type V MRS samples were isolated from veterinarians and the owners of dogs in the study area [10]. Recently, studies of SCC*mec* typing in cats and dogs from other geographic areas indicated the presence of type V and type VII SCC*mec* MRS in Asia. The type V SCC*mec* was the most commonly detected type in MRS in Europe and America [11,37,38,39,40].

## 5. Conclusions

MRS, including *S. aureus*, *S. intermedius* group, and coagulase-negative *Staphylococcus* with multidrug resistance phenotypes, was isolated from dogs with superficial pyoderma. These MRS infection findings pose certain diagnostic and treatment challenges for South East Asian veterinary practitioners and highlight the need for improved antimicrobial stewardship and hygienic practices. This includes diagnostic recognition and new treatment approaches. It also reminds veterinarians about the potential for zoonotic transmission of multidrug-resistant organisms. Veterinarians should consider appropriate treatment options, including effective topical treatments for mild cases to reserve the use of important parenteral treatments if needed. Newly released recommendations can serve as established approaches for the treatment of superficial pyoderma. The emergence of type V SCC*mec* MRS has broad implications for companion animals, pet owners, veterinarians, and animal caretakers. The emergence of these MRS strains can contaminate the hospital environment and are a public health concern.

## Figures and Tables

**Figure 1 vetsci-08-00085-f001:**
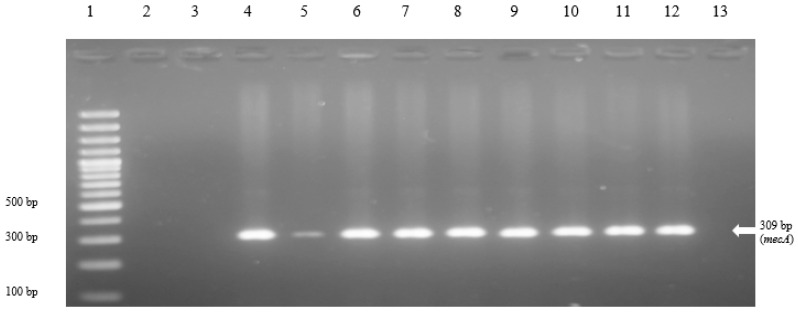
Results of *mecA* gene (309 bp) detection by the conventional PCR among 12 MRS strains. Lane 1, 100 bp plus marker; lane 2 (CMU1) and lane 3 (CMU2), samples with undetected *mecA* gene; lane 4 (CMU3), lane 5 (CMU68), lane 6 (CMU27), lane 7 (CMU29), lane 8 (CMU52), lane 9 (CMU62), lane 10 (CMU88), and lane 11 (CMU71), samples with detected *mecA* genes; lane 12 (CMU64) as a positive control for which *mecA* was detected using DNA sequence method; lane 13, negative control (distilled water).

**Figure 2 vetsci-08-00085-f002:**
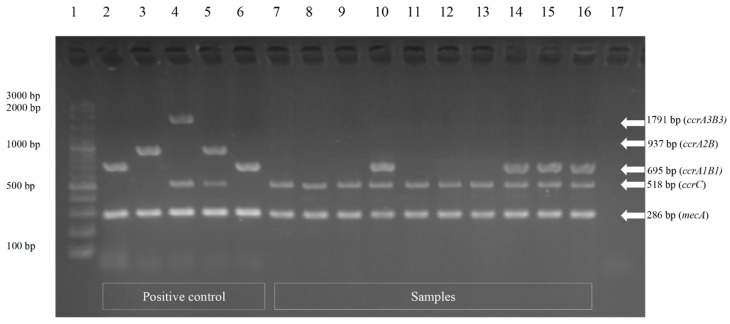
M-PCR1 for identification of *ccr* genes for assignment of the type of *ccr* gene complex. Lane 1, 100 bp plus marker; lane 2, positive control types I *ccrA1B1* (695 bp); lane 3, positive control type II *ccrA2B2* (937 bp); lane 4, positive control type III *ccr*A3B3 (1791 bp) and *ccrC* (518 bp); lane 5, positive control type IV *ccrA2B2* (937 bp) and *ccrC* (518 bp); lane 6, positive control type IX *ccrA1B1* (695 bp); lane 7 (CMU64), lane 8 (CMU3), lane 9 (CMU27), lane 11 (CMU29), lane 12 (CMU33), and lane 13 (CMU42), samples with *ccrC* (518 bp); lane 10 (CMU52), lane 14 (CMU62), lane 15 (CMU68), and lane 16 (CMU88), samples with *ccr*A1B1 (695 bp) and *ccrC* (518 bp); lane 17, negative control (distilled water). The 286 bp amplification products that appear in each lane represent *mecA*.

**Figure 3 vetsci-08-00085-f003:**
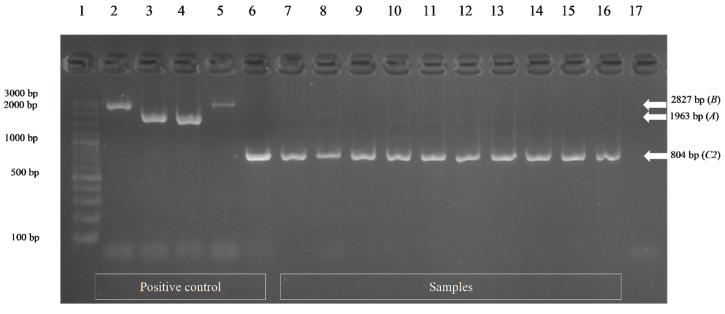
M-PCR2 for identification of three gene alleles for assignment of the *mec* gene complex. Lane 1, 100 bp plus marker; lane 2, positive control type I *mec* class B (2827) bp; lane 3, positive control type II *mec* class A (1963 bp); lane 4, positive control type III *mec* class A (1963 bp); lane 5, positive control type IV *mec* class B (2827 bp); lane 6, positive control type IX *mec* class C2 (804 bp); lane 7 (CMU64), lane 8 (CMU3), lane 9 (CMU27), lane 10 (CMU52), lane 11 (CMU29), lane 12 (CMU33), lane 13 (CMU42), lane 14 (CMU62), lane 15 (CMU68), and lane 16 (CMU88), sample *mec* class C2 (804 bp); lane 17, negative control (distilled water).

**Table 1 vetsci-08-00085-t001:** Antimicrobial susceptibility test result of 56 staphylococcal isolates from superficial pyoderma dogs.

Antimicrobial Agent	Antimicrobial Susceptibility *n* (%)		
	Susceptible	Intermediate	Resistant
Penicillin	19 (33.93%)	0 (0%)	37 (66.07%)
Ampicillin	21 (37.50%)	0 (0%)	35 (62.50%)
Cefoxitin	44 (78.57%)	0 (0%)	12 (21.43%)
Amoxicillin/clavulanate	43 (76.79%)	0 (0%)	13 (23.21%)
Cefazolin	43 (76.79%)	0 (0%)	13 (23.21%)
Cefpodoxime	43 (76.79%)	0 (0%)	13 (23.21%)
Amikacin	38 (67.86%)	15 (26.79%)	3 (5.36%)
Gentamicin	39 (69.64%)	14 (25%)	3 (5.36%)
Doxycycline	52 (92.86%)	4 (7.14%)	0 (0%)
Ciprofloxacin	44 (78.57%)	0 (0%)	12 (21.43%)
Chloramphenicol	46 (82.14%)	2 (3.57%)	8 (14.29%)
Clindamycin	38 (67.86%)	6 (10.71%)	12 (21.43%)
Erythromycin	37 (66.07%)	5 (8.93%)	14 (25%)
Linezolid	56 (100%)	0 (0%)	0 (0%)
Rifampin	56 (100%)	0 (0%)	0 (0%)
Trimethoprim/Sulfamethoxazole	38 (67.86%)	7 (12.50%)	17 (30.36%)

**Table 2 vetsci-08-00085-t002:** Results of methicillin-resistant staphylococci by oxacillin MIC, cefoxitin disk diffusion test, *mecA* gene detection, and SCC*mec* typing.

NO.	Isolate ID	Bacteria	Oxacillin MIC (µg/mL)	Cefoxitin (30 µg) Disk Diffusion Test	*mecA* Gene	*ccr* Gene Complex	*mec* Gene Complex	SCC*mec* Typing
1	CMU 3	*S. pseudintermedius*/*S*. *intermedius*	>4, Resistant	NA	Positive	*C*	C2	V
2	CMU 27	*S. pseudintermedius*/*S*. *intermedius*	>4, Resistant	NA	Positive	*C*	C2	V
3	CMU 29	*S. pseudintermedius/S. intermedius*	>4, Resistant	NA	Positive	*C*	C2	V
4	CMU 33	*Staphylococcus lentus*	0.5, Resistant	Resistant	Positive	*C*	C2	V
5	CMU 42	*Staphylococcus arlettae*	1, Resistant	Resistant	Positive	*C*	C2	V
6	CMU 52	*S. pseudintermedius/S. intermedius*	>4, Resistant	NA	Positive	*A1B1*, *C*	C2	Non-typeable
7	CMU 61	*Staphylococcus xylosus*	0.5, Resistant	Resistant	Positive	*C*	C2	V
8	CMU 62	*S. pseudintermedius/S. intermedius*	1, Resistant	NA	Positive	*A1B1*, *C*	C2	Non-typeable
9	CMU 64	*Staphylococcus aureus*	>4, Resistant	Resistant	Positive	*C*	C2	V
10	CMU 68	*S. pseudintermedius/S. intermedius*	1, Resistant	NA	Positive	*A1B1*, *C*	C2	Non-typeable
11	CMU 71	*S. pseudintermedius/S. intermedius*	>4, Resistant	NA	Positive	*C*	C2	V
12	CMU 88	*S. pseudintermedius/S. intermedius*	>4, Resistant	NA	Positive	*A1B1*, *C*	C	Non-typeable

NA, not applicable.

## Data Availability

The data presented in this study are available on request from the corresponding author. The data are not publicly available due to institutional privacy policy.

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
