# Peer review of "The SCCmec Types and Antimicrobial Resistance among Methicillin-Resistant Staphylococcus Species Isolated from Dogs with Superficial Pyoderma"

_vetsci, 2021, doi:10.3390/vetsci8050085_

Round 1
Reviewer 1 Report
Dear Authors,
you have submitted a very nicely presented manuscript of interesting topic. Nowdays, every research of MDR bacteria is important and contributes nicely to general knowledge on MDR. This manuscript also to some degree connects clinical importance and research of pathogens and this interpretation with respect to applicability of your results also adds additional value.
I have some minor suggestions/questions for you, as follows:
Line 31: I would find it clearer if the sentence was: Of the 65 clinical samples, 86.15% staphylococcal infections were identified, out of which 18.46% were MRS infections.
Figure 4 and 5: Why do you present results from 10 isolates if you had 12 isolates of MRS?
Line 241: It seems to me the sentence is more clear without the word "were".
Line 248: Clinical signs are caused by infection, not by detection of infection. Therefore, I would leave out "the detection of".
Line 262: you only tested for one fluoroquinolone (ciprofloxacin) so singular (fluoroquinolone instead of fluoroquinolones) might be more appropriate. Furthermore, all MRS isolates are resistant to penicilins and cephalosporines, so I don't find it necessary to report that for this study.
Line 269-270: I do not understand this sentence. What antimicrobials are recommended for treatment of MRS? Or did you mean treatment of staphylococcal infections in general? Please, make it more clear.
Line 283: "..in the existing culture." What did you mean? Veterinarians habits or bacterial culture? Please, make it more clear.
In Conclusions it may be nice to point out that veterinarians should try to adhere to current recommendations for treatment of pyoderma and use topical therapy whenever possible, especially if culture and AST results are not available.
Author Response
Dear Reviewer,
Thank you for your kind consideration on the Manuscript ID: vetsci-1147592 entitled " Staphylococcal Cassette Chromosome mec Types and Antimicrobial Resistance among Methicillin-Resistant Staphylococcus Species Isolated from Dogs with Superficial Pyoderma.
We went through the comments and revised the manuscripts as reviewer(s)’ comments which are major and minor comments. Therefore, English language edit was performed by native speaker, Prof. Dr. Jeff Bender. And we also corrected the data presentation, including figures and tables.
Sincerely

Reviewer 2 Report
Minor edits suggested in the manuscript:
line 96. Staphylococcal isolates were...
line 99. MALDI-TOF-MS
line 112. DNA was extracted using ...
Author Response

(The authors gave the same response as above.)

Reviewer 3 Report
Dear Authors,
Despite the article offering new ideas and a valid starting point for future work and research its development appears incomplete, above all because, there are very few data and in a very limited geographical area.
The information concerning the methodology, the data analysis, and the development of the results are clear and well defined. Particularly, the methodological approach foresees clear guidelines that can be useful for other researchers as a basis for future studies about the same aim.
Therefore, in my opinion, deep and substantial changes are still needed, so the manuscript in this form is not ready for publication on the Veterinary Sciences Journal.
Comments and Suggestions.
Line 1: I suggest the Authors to submit the manuscript as a Short Communication, since the data reported are few.
Lines 2-4: a shorter title would be appropriate, concise, and equally descriptive.
Line 38: control measures aimed at reducing environmental contamination and potential exposures to MRS are necessary not only in animal hospitals, but also includes veterinary hospital staff, clients, and patients.
Lines 40-41: the keywords are repetitive and should never be the same as indicated in the title. I suggest modifying them above all to increase the availability of the article.
Lines 51-52: In my opinion, it would be appropriate to move this sentence to line 47, where the authors are discussing Staphylococcus aureus.
Lines 82-85: This subsection is not needed in "Materials and Methods". The information reported should be moved to the relevant section after the conclusions because even if an animal experimentation was not carried out, skin swabs were performed on dogs involved in the study. Please change.
Line 107: Why was the antibiotic "cefoxitin" reported in a separate sentence from the others? Please clarify.
Lines 154-158: Please rephrase this sentence, as it is full of repetitions (ie isolated and identified) and above all the number "one" must be written in letter and not as "1".
Line 237 (Discussion): this section should be implemented with comparative data from the scientific literature. Several studies and reviews / meta-analyzes have addressed the zoonotic potential of Staphylococcus spp. methicillin resistant isolates from dogs. Furthermore, I suggest making this change also to give more interest and scientific solidity to the discussion.
Line 245: this finding highlight, also, the need to reduce the phenomenon of antibiotic resistance. Please integrate.
Line 261: I suggest changing drug with "multidrug resistant", it would be more correct.
Line 271: Please replace "observed" with "showed" is more appropriate.
After line 278: limits of the study should be included, since the dimension of the sample does not guarantee an adequate representation of the population, which interferes with the statistically significant interpretation of the results.
Line 288: not only about the contamination of the hospital environment, it concerns a public health problem. Please integrate.
Author Response

(The authors gave the same response as above.)

Reviewer 4 Report
Manuscript ID: vetsci-1147592
Type of manuscript: Article
Title: Staphylococcal Cassette Chromosome mec Types and Antimicrobial Resistance among Methicillin-Resistant Staphylococcus Species Isolated from Dogs with Superficial Pyoderma
This manuscript describes the characteristics of methicillin-resistant staphylococcal (MRS) isolates obtained from superficial pyoderma in dogs. It is well known, that the population of methicillin-resistant staphylococci in dogs is very diverse. Thus, results regarding strains isolated in different geographic regions seem very valuable for understanding the epidemiology of these bacteria.
However, I would suggest a minor revision before acceptance for publication. The recommendations to the author to improve the manuscript:
My most important comments concern the identification of staphylococci included in the SIG. First, MALDI-TOF should identify to the species level. In dogs, S. pseudintermedius is a particularly important and frequently isolated species. Recognition of this species is important due to separate recommendations used in the determination, different for S. aureus, others for S. pseudintermedius. The value of the manuscript would be much better if the authors provided a proper species identification of the SIG, which can also be achieved by simple PCR (described by Sasaki in 2010 - Multiplex-PCR Method for Species Identification of Coagulase-Positive Staphylococci). Moreover, two species have recently been added to the SIG - S. cornubiensis and S. ursi. Were they also present in the tested dogs? Currently, the concept of SIG is becoming more and more broad.
In my opinion, it is likely that most of the canine strains are S. pseudintermedius. According to the CLSI recommendations, an oxacillin disc (also for most of the CNS) should be used to detect methicillin resistance for this species. It is not clear to me why the authors only used a cefoxitin disc. Following the CLSI recommendations, oxacillin should be used. I suggest the authors to revise these results
Line 48: Methicillin resistance is found mostly in S. pseudintermedius, not in SIG in general.
Line 103: Recommendations cited by the authors (item number 13 in the list of literature) have been replaced by the CLSI document M100-S31 from 2021. Moreover, in case of strains of animal origin veterinary recommendations should be used, the current version.
Line 154: Did the authors isolate bacteria other than staphylococci from the clinical samples? What about 9 negative samples?
Line 250: Too general, especially from dogs S. pseudintermedius is isolated. Please see the general comment on new species in the SIG
I am convinced that these small corrections made by the authors will have a positive effect on the value of the manuscript.
Author Response

(The authors gave the same response as above.)
